# Role of LIN28B in the Regulation of Ribosomal Biogenesis and Lipid Metabolism in Medulloblastoma Brain Cancer Cells

**DOI:** 10.3390/proteomes13020014

**Published:** 2025-03-27

**Authors:** Ahmed Maklad, Mohammed Sedeeq, Kaveh Baghaei, Richard Wilson, John A. Heath, Nuri Gueven, Iman Azimi

**Affiliations:** 1School of Pharmacy and Pharmacology, College of Health and Medicine, University of Tasmania, Hobart, TAS 7001, Australia; ahmed.maklad@utas.edu.au (A.M.); mohammed.sedeeq@utas.edu.au (M.S.); nuri.guven@utas.edu.au (N.G.); 2Monash Biomedicine Discovery Institute, Department of Pharmacology, Monash University, Melbourne, VIC 3168, Australia; kaveh.baghaei@monash.edu; 3Central Science Laboratory, College of Science and Engineering, University of Tasmania, Hobart, TAS 7001, Australia; richard.wilson@utas.edu.au; 4Children’s Cancer Centre, Monash Children’s Hospital, Melbourne, VIC 3168, Australia; john.heath2@monashhealth.org

**Keywords:** brain cancer, LIN28B, lipid metabolism, medulloblastoma, ribosomal biogenesis

## Abstract

**Background:** Medulloblastoma (MB) is the most aggressive paediatric brain cancer, highlighting the urgent need for new diagnostic and prognostic biomarkers and improved treatments to enhance patient outcomes. Our previous study identified LIN28B, an RNA-binding protein, as a potential diagnostic and prognostic marker for MB and a pharmacological target to inhibit MB cell proliferation and stemness. However, the specific role of LIN28B and its mechanism of action in MB had not been studied. **Methods:** This study assessed LIN28B’s role in Daoy MB cells using siRNA-mediated silencing. LIN28B silencing was achieved with Dharmacon ON-TARGETplus SMARTpool and confirmed by Western blotting. Proliferation and protein assays evaluated the cell metabolic activity and viability. A proteomics analysis was conducted to examine the effect of LIN28B knockdown on the MB cell protein expression profile. The intracellular lipid droplets were assessed using the Nile Red Staining Kit, and nucleolar B23 protein levels were assessed by immunofluorescence. Both were visualised with a high-content IN Cell Analyser 2200. **Results:** Effective LIN28B silencing (>80%) was achieved in each experiment. LIN28B knockdown reduced the MB cell viability, impaired ribosome biogenesis, and promoted cellular lipid accumulation, as supported by proteomics and cell-based assays. **Conclusions:** This study highlights LIN28B as a promising target for regulating MB cell growth, ribosomal biogenesis, and lipid metabolism.

## 1. Introduction

Medulloblastoma (MB) is an embryonal brain tumour that occurs in the cerebellum [1,2]. It is considered the most common and fastest-growing primary malignant tumour in paediatric oncology [1] and accounts for around 20% of all paediatric brain tumours [3]. Standard treatments for MB include surgery, radiation therapy, and chemotherapy, which are typically used in combination to achieve better therapeutic results [1,4]. However, these strategies are not completely reliable and may not improve the prognosis, especially for high-risk MB [2,5]. Furthermore, current therapies are associated with serious adverse outcomes and additional health complications [5]. As a result, there is a critical need to identify new therapeutic approaches and targets to treat MB. One such new target might be LIN28B. Our previous study revealed a significant increase in the expression of LIN28B in MB tissues compared to non-tumour brain tissues [6]. This overexpression, which was correlated with lower patient survival and higher metastasis rates, was exclusive to MB and not shared with other brain cancers. This suggests that LIN28B could serve as a diagnostic and prognostic tool for MB [6]. In addition, we demonstrated that the pharmacological antagonist of LIN28 (Lin28-1632), targeting both LIN28A and LIN28B isoforms, suppresses MB cell viability and stemness [6]. These findings highlight the possibility that LIN28 could be used as a target for pharmacological interventions to impede MB cell growth and stemness. However, our previous study did not characterise the specific roles of the LIN28B isoforms in MB.

The LIN28 protein family includes two homologs, LIN28A (or LIN28) and its mammalian counterpart LIN28B, which are both evolutionarily conserved RNA-binding proteins [7,8,9]. LIN28A was originally identified in Caenorhabditis elegans as a heterochronic gene and shown to affect embryonic developmental timing, whereas LIN28B was first identified in hepatocellular carcinoma (HCC) [7,8,9,10]. Both proteins possess significant physiological roles in regulating pluripotency and tissue regeneration in embryonic stem cells (ESCs) [9,11,12,13,14]. Furthermore, the elevated expression of these proteins was used to diagnose and link to the poor prognosis of several malignant tumours of the nervous system, such as glioblastomas [15], atypical teratoid/rhabdoid tumours (AT/RTs) [16], primitive neuroectodermal tumours (PNETs) [17], embryonal tumours with multilayered rosettes (ETMRs) [16,18], neuroblastomas [19], and MBs [6]. Particularly, LIN28B expression is frequently elevated in neuroblastomas and medulloblastomas [6,19]. However, the specific role of the LIN28B protein in medulloblastoma remains unexplored.

Thus, by utilising siRNA-mediated silencing, proteomic analysis, and cell-based assays, this study aimed to identify the roles of LIN28B in MB and its potential as a target for future therapeutic approaches for MB patients.

## 2. Materials and Methods

### 2.1. Cell Culture

The Daoy MB cell line (ATCC^®^ HTB186™; belongs to SHH MB molecular subgroups) was purchased and cultured according to the American Type Culture Collection (ATCC; Manassas, VA, USA). Cells were grown in Eagle’s Minimum Essential Medium (#M0643; Sigma-Aldrich, St. Louis, MO, USA) with 10% foetal bovine serum (FBS). An incubator with a humidified atmosphere was used to keep the cells at 37 °C and 5% carbon dioxide. For the experiments, the cells were grown no more than ten passages.

### 2.2. siRNA-Mediated Knockdown Transfection

For silencing LIN28B, Dharmacon ON-TARGETplus SMARTpool LIN28B siRNA (L-028584-01-0005) and the DharmaFECT2 transfection reagents were used (T-2002-01; GE Healthcare Dharmacon, Lafayette, CO, USA), according to the manufacturer’s instructions. The ON-TARGETplus non-targeting pool (D-001810-10-05) was used as a control. For cell transfection, the cells were seeded at 0.5, 1 × 10^3^ cells per well in a clear DNase/RNase-free plate (TPP Techno Plastic Products™; # 92096; 96-well, flat-bottom plate). After 24 h post-plating, the cells were transfected with siLIN28B (50 nM per well), together with the DharmaFECT2 transfection reagent (0.1 µL per well). The effective knockdown levels were assessed 96 h post-siRNA treatment using immunoblotting.

### 2.3. Immunoblotting

Total protein extracts of Daoy cells and their concentrations were quantified and prepared for a Western blotting analysis using the Bio-Rad DC Protein Assay (catalogue #500-0116; Bio-Rad, Hercules, CA, USA), as previously published [6]. Protein samples of 5 µg were resolved on 10% Bis-glycine-polyacrylamide gel and then transferred to an Amersham^TM^ Protran^TM^ 0.2 µm nitrocellulose blotting membrane (catalogue #10600001; GE Healthcare Life Science, Chicago, IL, USA). Consequently, the membranes were blocked, and then the LIN28B protein was detected using a primary antibody against LIN28B (rabbit polyclonal; catalogue #HPA061745; Sigma-Aldrich, St. Louis, MO, USA; 1:1000 dilution) and against nucleophosmin (B23) (mouse monoclonal; catalogue #B0556; Sigma-Aldrich; 1:200 dilution). GAPDH or β-actin proteins were used as a loading control using an antibody against GAPDH (mouse monoclonal; catalogue #G8795; Sigma-Aldrich; 1:20,000 dilution) and β-actin (mouse monoclonal; catalogue #A5441; Sigma-Aldrich; 1:10,000 dilution). Goat anti-rabbit (catalogue #170-6515; Bio-Rad; 1:3000 dilution) or goat anti-mouse IgG (catalogue #170-6516; Bio-Rad; 1:3000 dilution) were used as secondary antibodies.

For visualising the immunoreactivity, an Amersham^TM^ ECLTM Prime Western blotting detection reagent (code #RPN2236; GE Healthcare Life Science) was used. A Chemi-Smart 5000 imager (Vilber Lourmat, Eberhardzell, Germany) was used to record digital images, and the Image Lab^TM^ software version 6.0.1 (Bio-Rad) was used to quantify the densities of the bands. The band intensities of the proteins of interest were normalised to the loading control band intensities for the determination of the relative protein expression levels. The protein band molecular weights (kDa) were determined using the PageRuler^TM^ Plus Prestained Protein Ladder (catalogue #26620; Thermo Scientific, Waltham, MA, USA). For the statistical comparison analysis, a two-tailed *t*-test with the Mann–Whitney test was used to analyse data by the GraphPad Prism software, version 9.1.

### 2.4. WST-1 Assay and Protein Content

WST-1 and protein assays were used to assess the impact of LIN28B silencing on the metabolic activity and viability of Daoy cells. The cells were seeded at 0.5 × 10^3^ cells/well in a 96-well plate for 24 h.

For the WST-1 colourimetric assay, 96 h post-transfection, 10 µL of the WST-1 mixture (#10008883, Cayman Chemical, Ann Arbor, MI, USA) was added to each well and incubated for 2 h. Post-incubation, the absorbance of each sample was measured using the Multiskan Go microplate reader (Thermo Fisher Scientific, Melbourne, VIC, Australia) at a wavelength of 450 nm, according to the manufacturer’s protocol. The absorbance signal was normalised to the protein content.

The protein content of the cell lysates was quantified using the Bio-Rad DC Protein Assay (colourimetric assay; catalogue #500-0116; Bio-Rad), according to the manufacturer’s protocol. The absorbance of each sample was measured using the Multiskan Go microplate reader (Thermo Fisher Scientific, Melbourne, VIC, Australia) at 750 nm.

### 2.5. Proteomic Analysis

#### 2.5.1. Protein Extraction and Sample Preparation

Three biological replicates, each in triplicate (18 samples in total) of two groups—the silenced LIN28B (siLIN28B) and the non-targeting (siNT) control—were prepared for a proteomics analysis. MB Daoy cells were seeded at 1 × 10^3^ cells per well in a 96-well plate. After 24 h post-plating, the cells were transfected with siLIN28B (50 μM) for 96 h. The cells were first washed with phosphate-buffered saline (PBS) and then lysed using a protein lysis buffer containing Tris base (50 mM; pH 8.0), Na deoxycholate (0.5%), IGEPAL^®^ (1%), NaCl (100 mM), 1 × phosphatase, and protease inhibitors (Roche Diagnostic GmbH, Mannheim, Germany). As aforementioned, the Bio-Rad DC Protein Assay was used to measure the protein concentrations.

#### 2.5.2. Protein Reduction, Alkylation, and Digestion

To prepare protein samples for a liquid chromatography–mass spectrometry (LC-MS/MS) analysis, 30 µg protein aliquots were progressively reduced using 10 mM dithiothreitol (DTT; overnight at 4 °C) and alkylated with 50 mM iodoacetamide (2 h in the dark at room temperature). Trypsin + rLysC mix (20 µg; Promega Madison, WI, USA) was reconstituted in 100 mM ammonium bicarbonate, and the samples were digested overnight according to the single-pot, solid-phase-enhanced sample-preparation (SP3) protocol [20]. Digestion was terminated with 0.1% trifluoroacetic acid (TFA), followed by centrifugation for 20 min at 21,000× *g* and peptide desalting with ZipTips (Merck, Darmstadt, Germany), as per the manufacturer’s instructions.

#### 2.5.3. Mass Spectrometry and Proteomics Data Analysis

The peptide samples were analysed using the UltiMate 3000 RSLCnano high-performance liquid chromatography (HPLC) system coupled to a Q Exactive™ HF Hybrid Quadrupole-Orbitrap™ MS (QE-HF) system (Thermo Scientific, Waltham, MA, USA). The samples were separated over a 120-min gradient with the MS system operated in the data-independent acquisition (DIA) mode, as described in Chear et al. [21].

The 18 DIA-MS raw files were converted to an HTRMS format, and the Pulsar search engine was then used to search the MS/MS spectra against the UniProt reference proteome database for Homo sapiens (comprising 20,443 sequences, one entry per gene entry) to generate a spectral library using the Spectronaut software v.14.8 (Biognosys AG, Zurich, CH, Switzerland). The library, comprising 50,556 prototypic peptides, was then used for targeted re-extraction of the MS/MS spectra according to the default Biognosys (BGS) settings for label-free quantitation (LFQ) and cross-run normalisation, with the exception that single-hit proteins were excluded. The complete set of proteins identified and their respective LFQ intensity values across the samples are reported in Appendix A. The MS proteomics data have been deposited to the ProteomeXchange Consortium via the PRIDE [22] partner repository with the dataset identifier PXD022481, with the following login credentials: Username: reviewer_pxd022481@ebi.ac.uk; Password: i2lMfLQV (raw files AM1-9 map to control samples, and raw files AM10-18 map to the siLIN28B samples).

The proteins whose relative abundance was significantly altered in abundance in the siLIN28B samples relative to non-targeting control samples were identified using a two-tailed *t*-test with an FDR adjustment for multiple hypothesis testing (q < 0.05; threshold [+/−] 1.5-fold change) and displayed as a volcano plot using the Perseus software (v.1.6.50), the results of which are reported in Appendix A. These proteins were then submitted to two online bioinformatics resources, the PANTHER online tool (v.17.0) with a GO biological process annotation [23] and the STRING database (v.11.0b; https://version-11-0b.string-db.org/; accessed on 10 December 2020) [24] in order to determine functional significance and biological pathways for protein–protein interactions (Appendix A).

### 2.6. Nucleolar Immunofluorescence Analysis

Three biological replicates, each in triplicate of two groups—siLIN28B and the NT control—were prepared for immunofluorescence staining. Daoy cells were seeded at 1 × 10^3^ cells/well in black 96-well plates (#655090; µClear^®^). After 24 h post-seeding, the cells were transfected with siNT or siLIN28B. After 96 h, the cells were fixed for 10 min with formaldehyde (4%) and then permeabilised for another 10 min by 0.5% Triton^®^-X diluted in PBS. After being blocked for 1 h with 10% normal goat serum (#G9023; Sigma-Aldrich, St. Louis, MO, USA) in 0.1% Tween-20 in PBS (PBST), the cells were stained and exposed to a primary antibody against B23 (mouse monoclonal; FC82291; catalogue #B0556; Sigma-Aldrich; 1:200 dilution) and incubated overnight at 4 °C. After the incubation, a fluorescently tagged secondary antibody (F(ab’)2-goat anti-mouse IgG (H + L), Alexa Fluor™ 647; catalogue #A-21237; 1:1000 dilution) was added to the cells for 1 h at room temperature. For the nuclear staining, we used 4′,6-diamidino-2-phenylindole (DAPI) dye (1 µM; 1:10,000 dilution) for a 10 min incubation at room temperature. A high-content cell imaging instrument, the IN Cell Analyser 2200 (GE Healthcare Life Sciences, Marlborough, MA, USA), was used to capture and record the images at 60× magnification. The IN Carta Image Analysis Software, version 1.15 (GE Healthcare Life Sciences, Marlborough, MA, USA) was utilised to automatically quantify the B23 fluorescence intensity.

For a nucleolar analysis, a total of nine distinct fields across three wells of each group (siLIN28B or siNT) were imaged from three independent experiments. The IN Carta Image Analysis Software (GE Healthcare Life Sciences, Marlborough, MA, USA) was used to analyse the data. Overall, 1204 cells for the siLIN28B group and 1075 cells for the siNT group were analysed.

### 2.7. Assessment of Intracellular Lipid Droplets

siLIN28B and the NT control were prepared for assessing intracellular lipid droplets. The assessment of intracellular lipid droplets was demonstrated using the Nile Red Staining Kit (ab228553; Abcam, Cambridge, UK), according to the manufacturer’s instructions. MB Daoy cells were seeded at 1 × 10^3^ cells per well in black 96-well plates (655090; µClear^®^; Greiner Bio-One, Kremsmünster, Austria). After 24 h post-plating, the cells were transfected with siLIN28B (50 μM) or siNT (50 μM) for 96 h. The cells were washed twice in warm PBS (100 µL) and then incubated with the assay solution containing the Nile Red dye (3 μM; 50 µL) for 10 min at 37 °C/5% CO_2_. Consequently, the cells were washed one time and resuspended in warm PBS (100 µL). Immediately, the cells were imaged using the IN Cell Analyser 2000 (fluorescence at 550/640 nm; 37 °C) at 60× magnification. The IN Carta Image Analysis Software (GE Healthcare, Marlborough, MA, USA) was used to automatically quantify the total fluorescence of lipid droplets for each cell of each group. The Nile Red dye (Cy3; Gold) was used to stain the lipid droplets, while Hoechst DNA staining (blue) was used for nuclear staining. Twelve different fields/wells of three wells of each group (siLIN28B vs. siNT) were imaged from two independent experiments. Overall, 3445 cells for the siLIN28B group and 5236 cells for the NT group were analysed.

### 2.8. Statistical Analysis

The data analysis for the proteomics is described in detail above (Section 2.5.3). The GraphPad Prism statistical analysis software was used for all analyses (version 9.1 for Windows). A two-tailed unpaired *t*-test was used for a parametric statistical analysis (to compare siLIN28B and the NT control in the Daoy cells). The data were presented as means ± SDs, and a value of *p* < 0.05 was regarded to indicate statistical significance. The significance of each experiment and the statistical tests used are described in the accompanying figure legends.

## 3. Results

### 3.1. LIN28B Silencing Reduces Viability of MB Cells

To assess if LIN28B is important for the viability of the MB cells, we first determined the effect of siRNA-mediated silencing of LIN28B (siLIN28B) on the viability of the Daoy MB cells. Effective LIN28B siRNA-mediated silencing greater than 80% for each experiment was confirmed by Western blotting (Figure 1A). The reduced LIN28B expression significantly reduced the cell viability by about 25%, quantified both by the relative protein content per well (Figure 1B) and by the WST-1 dye conversion (Figure 1C).

### 3.2. LIN28B Knockdown Regulates Expression of Proteins Associated with RNA Biogenesis and Different Metabolic Pathways

To further understand the role of LIN28B in MB, we used label-free proteomics to study the effect of LIN28B knockdown on the protein expression profile of MB cells, resulting in a dataset comprising > 4800 proteins (excluding single-peptide protein IDs) (Appendix A). Using a *t*-test analysis, 295 proteins met the criteria for statistical significance with an FDR < 0.05 and a fold change > 1.5, of which 167 were significantly increased in abundance in the LIN28B knockdown cells, while 128 proteins were significantly decreased in abundance (Figure 2A; Appendix A). Using the PANTHER (v.17.0) and STRING (v.11.0b) databases, several distinct pathways were found to be associated with these two sets of significant proteins (Appendix A). For the proteins downregulated by the LIN28B siRNA treatment, many of the significantly enriched functional terms were associated with rRNA processing and ribosome biogenesis, while proteins that were upregulated by the LIN28B siRNA treatment were linked to cellular, lipid-, cellular catabolic-, and small molecule metabolic processes (Appendix A). The STRING analysis identified enriched biological processes that were highly consistent with the PANTHER analysis when considering the decreased proteins in abundance (Appendix A). For the proteins detected at higher levels, in contrast to the STRING analysis, the oxidation–reduction pathway was not enriched in the PANTHER analysis (Appendix A). The STRING software (v.11.0b) was then used for a functional enrichment visualisation (Figure 2B,D) and a protein–protein interaction network analysis (Figure 2C,E) of the sets of significant proteins. Based on the combined results of the STRING and PANTHER analyses, we opted to focus on ribosomal biogenesis to represent the set of downregulated proteins and lipid metabolism to represent the proteins upregulated by the siLIN28B treatment.

### 3.3. LIN28B Promotes Expression of Proteins Involved in Ribosomal Biogenesis

Ribosomal biogenesis primarily takes place in the nucleolus [25,26]. Thus, to examine the effect of LIN28B silencing on ribosomal biogenesis, we assessed nucleolar size in addition to the expression of the nucleolar phosphoprotein B23 (nucleophosmin), which is involved in ribosome biogenesis [27,28,29]. This was achieved via two approaches: the first approach was to examine the size of cell nucleoli, which is an indication of ribosomal biogenesis and cell proliferation [30,31,32]. Using B23 to label nucleoli as a method of nucleoli labelling [33], we showed that nucleolar size was significantly reduced in LIN28B-silenced Daoy cells compared to the siNT cells (Figure 3A,B), while the number of nucleoli in both groups was not significantly different (Figure 3C). Similarly, LIN28B knockdown in Daoy cells significantly inhibited B23 expression by about 50% (Figure 3D). Collectively, these data demonstrate that LIN28B silencing reduces the nucleolar size and B23 expression, suggesting a decrease in ribosomal biogenesis.

### 3.4. LIN28B Inhibits Lipid Accumulation in Daoy Cells

Given the observed proteins detected at higher levels in the LIN28B siRNA treatment related to lipid metabolism (Figure 2), we assessed the effect of LIN28B silencing on lipid metabolism by measuring intracellular lipid content (Figure 4A). Daoy cells with downregulated LIN28B (siLIN28B) showed significantly higher levels of Nile Red fluorescence intensity (equivalent to increased cellular content of lipid droplets) compared to Daoy cells that expressed normal levels of LIN28B (siNT) (Figure 4B).

## 4. Discussion

We previously showed that the pharmacological suppression of LIN28 proteins (using Lin28-1632) in MB cells led to a reduction in LIN28B protein, inhibition of cell viability, and stemness [6]. In addition, LIN28B gene expression was highly upregulated in tissues from MB patients compared to non-tumour brain tissues, where its higher expression was correlated with a higher metastasis rate and a lower patient survival rate [6]. In this study, we assessed the role of LIN28B using siRNA-mediated silencing of LIN28B in Daoy MB cells. LIN28B silencing significantly reduced cell viability, suggesting the importance of this LIN28 isoform for MB cells. The link between targeting LIN28B and its involvement in tumour initiation, stemness, growth, metabolism, resistance to therapy, recurrence, and metastasis has been widely reported for other cancer types but not for MB [34,35,36,37,38,39].

This study used a proteomics approach to test the impact of LIN28B silencing on other biological processes in Daoy MB cells. It should be noted that our proteomic analysis focused at the level of pathways and protein interaction networks rather than perturbation of specific proteoforms, as the reference proteome database used to search the MS/MS data did not include non-canonical proteoforms, such as splice variants. This analysis demonstrated that knockdown of LIN28B identified a significant number of proteins involved in ribosome biogenesis. Ribosomal biogenesis is a complex process that involves the transcription, processing, and assembly of ribosomal RNA (rRNA) and ribosomal proteins to form ribosomes, the cellular organelles responsible for protein synthesis [40]. Previous studies identified LIN28 as a regulator of ribosome biogenesis in murine neural progenitor and human cells [33,41]. In addition to localising to the cytoplasm and nucleus, LIN28B can also localise in the nucleolus [7,42], which is where ribosomal biogenesis occurs [25,26]. It was previously reported that ribosomal biogenesis proportionally correlates with nucleolar size [30,32]. Although direct evidence of LIN28B’s involvement in rRNA processing in cancer is limited, it has been reported that LIN28B enhances ribosomal RNA biogenesis and ribosome function in MYCN-amplified neuroblastomas. Missios et al. showed that LIN28B utilises RNA-binding domains to preferentially interact with MYCN-induced transcripts, enhancing their translation. This links the MYCN-driven transcriptional program to increased ribosomal translation, driving the metastatic phenotype in cancer cells [37]. Our results showed a significant reduction in nucleolar area and B23 protein expression in Daoy cells treated with LIN28B siRNA compared to cells treated with the siNT control. However, we cannot exclude the possibility that reduced nucleolar size is associated with a reduced proliferative capacity of the cells, as described previously [31]. Taken together, these results confirmed that LIN28B promotes ribosomal biogenesis and nucleolar size in MB cells.

Our bioinformatic analysis, using PANTHER (v.17.0) and STRING (v.11.0b) tools, showed that the enriched biological processes were highly consistent between the two tools for downregulated proteins in the LIN28B siRNA treatment. However, for the proteins detected at higher levels, the oxidation–reduction pathway was not enriched in the PANTHER analysis, while it was identified by the STRING analysis. The differences in algorithms and data sources used by the two tools might contribute to these discrepancies. PANTHER provides curated protein families and subfamilies and encompasses a broader range of gene ontology categories, such as molecular functions and biological processes [43,44]. On the other hand, STRING uses a score-based method to measure the strength of protein–protein interactions and generates protein–protein interaction networks based on experimental evidence and computational predictions [24,45]. Additionally, each tool has its own strengths and limitations, which might affect the interpretation of the results. Nevertheless, our proteomics data in both datasets consistently demonstrated the involvement of the proteins detected at higher levels associated with LIN28B knockdown in multiple pathways required for lipid metabolism. Lipid metabolism in both normal and malignant cells was previously reported to be controlled by LIN28A, LIN28B, and the pharmacological suppression of LIN28 [46,47]. LIN28B has been shown to regulate lipid metabolism by targeting specific mRNAs involved in lipid synthesis and storage, thereby controlling the balance of fatty acid and lipid storage in cells. For example, LIN28A and LIN28B regulate the expression of genes, such as the transcription factor SREBP1, which is involved in the regulation of fatty acid synthesis, thus enhancing lipid accumulation and de novo fatty acid synthesis in liver cancer cells [47]. However, our results showed that this is not the case in Daoy MB cells, as the knockdown of LIN28B promoted lipid accumulation. This suggests that the effect of LIN28A and LIN28B on lipid metabolism may be cell-type specific and might also depend on the specific metabolic state of the cell. In the case of Daoy MB cells, knockdown of LIN28B may have led to a compensatory increase in lipid synthesis and accumulation, possibly as a response to other metabolic stresses. It is also important to note that the precise mechanisms by which LIN28B regulates lipid accumulation in normal and cancer cells are still not fully understood. Overall, this observation supported our proteomics results and suggested that LIN28B regulates lipid metabolism in MB cells. However, additional validation beyond Nile Red fluorescence, such as triglyceride (TG) quantification or BODIPY staining, would further strengthen these findings. Future studies should incorporate multiple lipid quantification techniques to provide a more comprehensive assessment of LIN28B’s role in lipid metabolism in medulloblastomas.

Beyond its role in MB, LIN28B has been implicated in several other malignancies, including hepatocellular carcinoma, colorectal cancer, and ovarian cancer, where it contributes to tumour progression, stemness, and resistance to therapy [35,48,49,50,51,52]. Its regulatory influence on Let-7 and other oncogenic pathways positions LIN28B as a potential biomarker for prognosis and a promising therapeutic target [42,53,54,55,56]. Efforts to develop LIN28B inhibitors are ongoing [57,58], highlighting its translational relevance across multiple cancer types.

## 5. Conclusions

In conclusion, this study highlights the possibility of inhibiting LIN28B using siRNA to suppress MB cell growth and ribosome biogenesis, which also promotes cellular lipid accumulation. Despite these findings, there is still one main limitation to our study. Our experiments were performed on one MB cell line (Daoy) belonging to the SHH subgroup. Therefore, it is scientifically difficult to extend our results to specific MB subgroups or to MB in general. Hence, further studies are warranted to determine and extend our observations to other MB subgroups to check whether our findings can be generalised or only are associated with specific groups. If the findings are only associated with specific groups, could that be exploited for group-specific therapeutic approaches?

## Figures and Tables

**Figure 1 proteomes-13-00014-f001:**
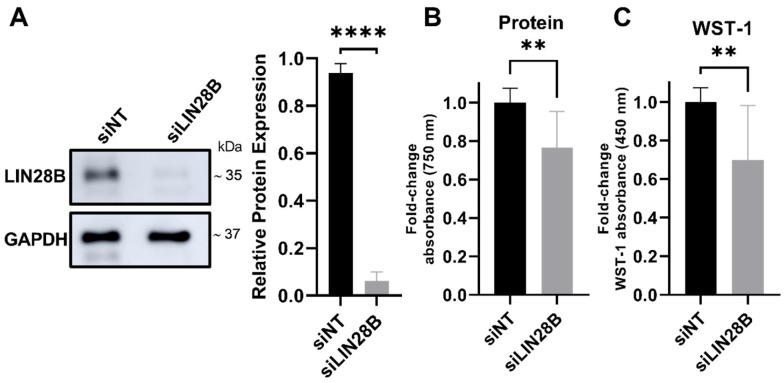
LIN28B knockdown inhibits cell viability of Daoy MB cells. (**A**) Representative immunoblot (**left**) and densitometric analysis (**right**) of LIN28B protein levels normalised to GAPDH as a loading control. (**B**) Protein content and (**C**) WST-1 dye conversion were measured for cells expressing physiological levels of LIN28B (siNT) versus cells with downregulated LIN28B levels (siLIN28B). siLIN28B data were normalised as fold change to the siNT control. Data represent mean ± SD from three independent experiments. **** *p*  <  0.0001, ** *p* < 0.01, two-tailed unpaired *t*-test.

**Figure 2 proteomes-13-00014-f002:**
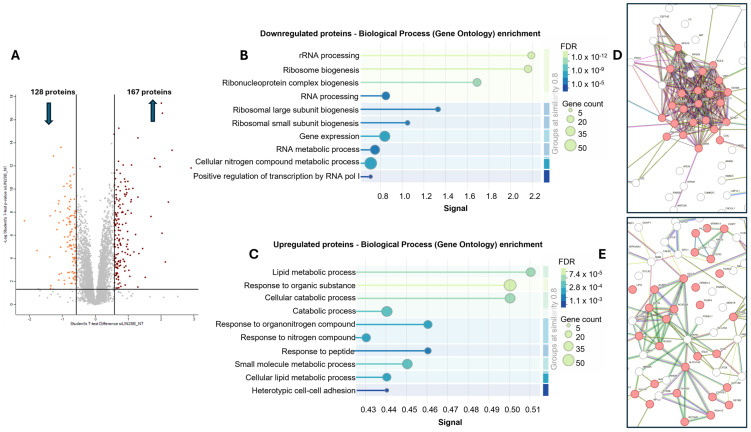
Overview of the proteomic and bioinformatic analysis used to investigate the effect of LIN28B knockdown in MB cells. Volcano plot displaying the result of a *t*-test analysis (**A**). Functional enrichment visualisation of biological processes associated with the proteins significantly downregulated (**B**) or upregulated (**C**) by the siLIN28B transfection of Daoy MB cells using the STRING software. Protein interaction networks based on the significantly downregulated (**D**) or upregulated (**E**) proteins, in which proteins related to “rRNA processing” and “lipid metabolic process” are highlighted in red. Only the network regions with highlighted proteins are shown. The full lists of significantly enriched biological processes from STRING v.11.0b and PANTHER v.17.0 [23] analyses are shown in Appendix A.

**Figure 3 proteomes-13-00014-f003:**
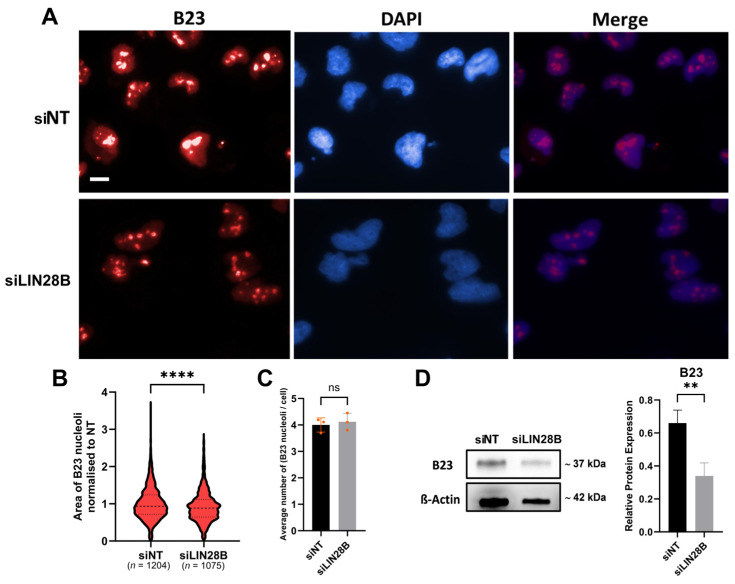
Effect of LIN28B on nucleoli size and number, and B23 expression in Daoy MB cells. (**A**) Representative immunofluorescence images of MB Daoy cells transfected with siRNA against LIN28B or siNT, stained against B23 (red) and DAPI DNA staining (blue), 60× magnification, scale bar, 20 μm. (**B**) Violin plot showing quantitative data of B23 nucleolar area. Data of siLIN28B-treated cells were normalised as fold change to siNT control. (**C**) Quantification of average nucleoli per cell. Data represent means ± SDs from a single-cell analysis of a total of 2279 cells, encompassing 1204 cells treated with siNT and 1075 cells treated with siLIN28B across three independent experiments. (**D**) Representative immunoblot (**left**) and densitometry analysis of total B23 protein levels normalised to β-actin as loading control. ns: not significant, ** *p* < 0.01, **** *p* < 0.0001, two-tailed unpaired *t*-test. n, represents the number of cells used in the analysis.

**Figure 4 proteomes-13-00014-f004:**
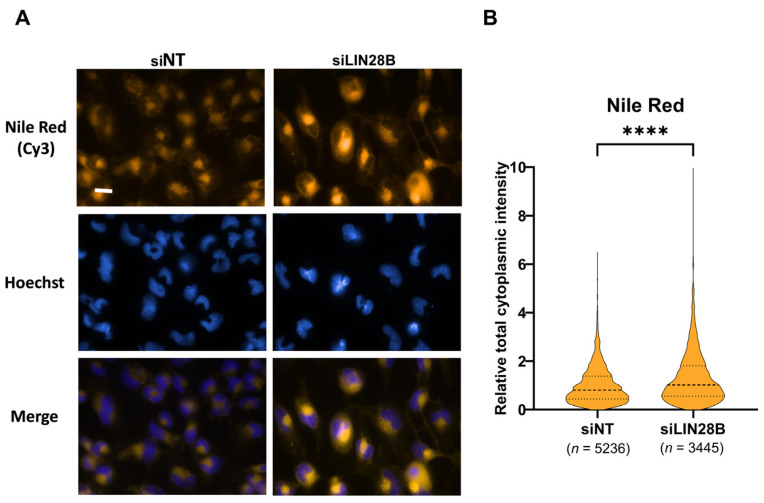
Effect of LIN28B knockdown on lipid accumulation in MB cells. (**A**) Representative fluorescence images of Daoy cells stained with Nile Red (Gold) and Hoechst (blue). 60× magnification, scale bar = 20 μm. (**B**) Violin plot showing quantitative data of cytoplasmic total intensity of lipid droplets of Daoy cells. Data of siLIN28B were normalised as fold change to siNT control. Data represent means ± SDs from a single-cell analysis of a total of 8681 cells, encompassing 5236 cells treated with siNT and 3445 cells treated with siLIN28B. **** *p* < 0.0001, two-tailed unpaired *t*-test. n represents the number of cells used in the analysis.

## Data Availability

The MS proteomics data have been deposited to the ProteomeXchange Consortium via the PRIDE partner repository with the dataset identifier PXD022481 with the following login credentials: Username: reviewer_pxd022481@ebi.ac.uk; Password: i2lMfLQV.

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
