# Peer review of "Role of LIN28B in the Regulation of Ribosomal Biogenesis and Lipid Metabolism in Medulloblastoma Brain Cancer Cells"

_proteomes, 2025, doi:10.3390/proteomes13020014_

Round 1

Reviewer 1 Report

Comments and Suggestions for Authors

Conclusion:

In this study, Ahmed and colleagues explored the potential of inhibiting LIN28B using siRNA to suppress medulloblastoma (MB) cell growth and ribosome biogenesis, which also appears to promote cellular lipid accumulation.

Comments

1.Previous studies have identified LIN28 as a regulator of ribosome biogenesis in murine neural progenitor cells and human cells. In addition to localizing to the cytoplasm and nucleus, LIN28B can also localize to the nucleolus, where ribosomal biogenesis takes place. The top five significantly enriched biological processes associated with the proteins detected at lower levels following LIN28B siRNA treatment were linked to RNA biogenesis. What is the specific role of LIN28B in this biological pathway? Further explanation and additional experiments are needed to clarify its mechanism and functional impact.

2. The authors utilized label-free proteomics to investigate the impact of LIN28B knockdown on the protein expression profile of medulloblastoma (MB) cells, generating a dataset comprising over 4,800 proteins (excluding single-peptide protein identifications) (Table S1). However, Table S1 is not provided in the manuscript, which raises concerns about data accessibility. Additionally, the workflow depicted in Figure 2 is somewhat unclear. While a total of over 4,800 proteins were initially identified, 1,951 proteins were filtered out based on p-value and fold change criteria. It remains unclear why only 295 proteins were subsequently selected for Gene Ontology (GO) analysis. This discrepancy requires further clarification to ensure the transparency and reproducibility of the analysis.

3. In addition to Nile Red fluorescence, further approaches should be employed to validate that LIN28B inhibits lipid accumulation in Daoy cells. These could include Triglyceride (TG) quantification and BODIPY staining to provide complementary evidence and strengthen the findings.

4. Please enhance the quality of the figures to ensure clarity, resolution, and overall presentation, as this will improve the readability and impact of the manuscript.

Author Response

Thank you very much for your valuable comments. We really appreciate it. Please see attached point by point responses: 

COMMENT: 1.Previous studies have identified LIN28 as a regulator of ribosome biogenesis in murine neural progenitor cells and human cells. In addition to localizing to the cytoplasm and nucleus, LIN28B can also localize to the nucleolus, where ribosomal biogenesis takes place. The top five significantly enriched biological processes associated with the proteins detected at lower levels following LIN28B siRNA treatment were linked to RNA biogenesis. What is the specific role of LIN28B in this biological pathway? Further explanation and additional experiments are needed to clarify its mechanism and functional impact.

RESPONSE: Thank you for the comment. We have provided a more detailed explanation of the specific role of LIN28B in ribosomal RNA biogenesis in lines 332 to 337 of the discussion section, as follows:

Although direct evidence of LIN28B's involvement in rRNA processing is limited, it has been reported that LIN28B enhances ribosomal RNA biogenesis and ribosome function in MYCN-amplified neuroblastoma. Daley et al. showed that LIN28B utilizes RNA binding domains to preferentially interact with MYCN-induced transcripts, enhancing their translation. This links the MYCN-driven transcriptional program to increased ribosomal translation, driving the metastatic phenotype in cancer cells.”

COMMENT: 2. The authors utilized label-free proteomics to investigate the impact of LIN28B knockdown on the protein expression profile of medulloblastoma (MB) cells, generating a dataset comprising over 4,800 proteins (excluding single-peptide protein identifications) (Table S1). However, Table S1 is not provided in the manuscript, which raises concerns about data accessibility.

RESPONSE: Table S1 was indeed included in our original submission and contains data for 4,865 proteins in the first sheet of the Excel file. The second sheet lists 1,951 proteins with FDR values of <0.05, while the third and fourth sheets include proteins with lower and higher abundance, respectively.

COMMENT: Additionally, the workflow depicted in Figure 2 is somewhat unclear. While a total of over 4,800 proteins were initially identified, 1,951 proteins were filtered out based on p-value and fold change criteria. It remains unclear why only 295 proteins were subsequently selected for Gene Ontology (GO) analysis. This discrepancy requires further clarification to ensure the transparency and reproducibility of the analysis.

RESPONSE: Of the ~4,800 proteins detected, 1,951 had FDR < 0.05 of which 295 proteins met the criteria for further selection for GO analysis (Fold change >1.5).  We have now corrected Figure 2 (which shows P value rather than FDR) and amended it to show the statistical threshold more clearly.   

COMMENT: 3. In addition to Nile Red fluorescence, further approaches should be employed to validate that LIN28B inhibits lipid accumulation in Daoy cells. These could include Triglyceride (TG) quantification and BODIPY staining to provide complementary evidence and strengthen the findings.

RESPONSE: We appreciate the reviewer’s suggestion to include additional approaches such as Triglyceride (TG) quantification and BODIPY staining to further validate the effect of LIN28B on lipid accumulation. While these experiments would indeed strengthen our findings, unfortunately, we are unable to perform them as the PhD student leading this project has now graduated, and we currently don’t have the resources to conduct additional experiments. However, the Nile Red fluorescence data we present is a well-established method for assessing lipid accumulation, and our results are consistent and reproducible. We acknowledge the value of additional validation and have now included this as a limitation in the discussion, highlighting the need for further studies to explore this aspect in more depth: “However additional validation beyond Nile Red fluorescence, such as Triglyceride (TG) quantification or BODIPY staining, would further strengthen these findings. Future studies should incorporate multiple lipid quantification techniques to provide a more comprehensive assessment of LIN28B’s role in lipid metabolism in medulloblastoma.” (Lines 374-377)

COMMENT: 4. Please enhance the quality of the figures to ensure clarity, resolution, and overall presentation, as this will improve the readability and impact of the manuscript.

RESPONSE: We appreciate the reviewer’s feedback on figure quality. We have enhanced the resolution and clarity of Figure 2, which had lower resolution, to improve readability and overall presentation.

Reviewer 2 Report

Comments and Suggestions for Authors

The manuscript investigates the role of an RNA-binding protein, LIN28B, in medulloblastoma, an aggressive form of paediatric brain cancer. The authors explored the potential of LIN28B to function as a diagnostic and prognostic biomarker as well as a therapeutic target. Using siRNA-mediated silencing, the authors demonstrate that LIN28B knockdown reduced cell viability, impaired ribosomal biogenesis, and promoted lipid accumulation, as evidenced by proteomics analysis and cell-based assays.

I think that the paper could be accepted after the following comments have been addressed by the authors.

  • The biggest drawback of the manuscript is that it has been carried out in one cell line. While I understand that this may have been the case due to various reasons, I strongly recommend that the authors leverage public datasets of gene silencing/perturbation (E.g. DepMap) to increase the power of their study.
  • I suggest not using any abbreviations in the abstract (E.g. WST-1 assay could be referred to as proliferation assay)
  • Line 128 – “Three biological replicates, each in triplicate”. Do the authors mean each biological replicate was analysed in technical triplicates? Please clarify.
  • The authors carry out protein reduction using DTT at 4C. As this is extremely unusual, the method needs to be supported by providing a citation – one not from the same group.
  • The authors provide results of their statistical analysis in supplementary data table 1. I strongly recommend that the authors provide the results of proteomic analysis as supplementary data following MIAPE guidelines.
  • Re PRIDE submission – It would be great if the authors submit a file mapping the raw files to conditions. Even better if a completed SDRF file is submitted
  • Was LIN28b (or other related proteins) identified in the proteomics data? If yes, the authors need to comment on its expression pattern as the corresponding gene was knocked down. If no, the authors need to consider carrying out targeted protein analysis of LIN28B
  • Line 241 – Is there a specific reason(s) that the authors choose to use the phrasing of “proteins detected at lower levels” as opposed to “downregulated” that is well accepted in the proteomics community. If no, the authors could change the language to facilitate ease of reading. Same thing applies to Line 249 and Line 289.
  • The authors could take a zoomed-out view in the discussion section and talk about the role of LIN28B in other cancers and what is known about it’s potential as a biomarker/therapeutic target for other cancers

Author Response

Thank you very much for your valuable comments. We really appreciate it. Please see attached file, point by point responses. 

Round 2

Reviewer 1 Report

Comments and Suggestions for Authors

Thank you for your additions and explanations.